# Therapeutic Drug Monitoring in Children and Adolescents: Findings on Fluoxetine from the TDM-VIGIL Trial

**DOI:** 10.3390/pharmaceutics15092202

**Published:** 2023-08-25

**Authors:** Michael Frey, Lukasz Smigielski, Elvira Tini, Stefanie Fekete, Christian Fleischhaker, Christoph Wewetzer, Andreas Karwautz, Christoph U. Correll, Manfred Gerlach, Regina Taurines, Paul L. Plener, Uwe Malzahn, Selina Kornbichler, Laura Weninger, Matthias Brockhaus, Su-Yin Reuter-Dang, Karl Reitzle, Hans Rock, Hartmut Imgart, Peter Heuschmann, Stefan Unterecker, Wolfgang Briegel, Tobias Banaschewski, Jörg M. Fegert, Tobias Hellenschmidt, Michael Kaess, Michael Kölch, Tobias Renner, Christian Rexroth, Susanne Walitza, Gerd Schulte-Körne, Marcel Romanos, Karin Maria Egberts

**Affiliations:** 1Faculty of Applied Healthcare Science, Deggendorf Institute of Technology, 94469 Deggendorf, Germany; 2Department of Child and Adolescent Psychiatry, Psychosomatics and Psychotherapy, University Hospital, LMU Munich, 80097 Munich, Germany; 3Department of Child and Adolescent Psychiatry and Psychotherapy, Psychiatric University Hospital Zurich, University of Zurich, 8032 Zürich, Switzerland; lukasz.smigielski@kjpd.uzh.ch (L.S.);; 4Department of Child and Adolescent Psychiatry, Psychosomatics and Psychotherapy, Center for Mental Health, University Hospital of Wuerzburg, 97080 Wuerzburg, Germany; 5Department of Child and Adolescent Psychiatry and Psychotherapy, University Medical Center Freiburg, 79104 Freiburg, Germany; 6KIRINUS Tagesklinik Kinder und Jugendliche, 80639 Munich, Germany; 7Department of Child and Adolescent Psychiatry, Medical University Vienna, 1090 Vienna, Austria; 8Department of Child and Adolescent Psychiatry, Charité Universitätsmedizin Berlin, 13353 Berlin, Germany; 9Department of Psychiatry, The Zucker Hillside Hospital, Northwell Health, Glen Oaks, NY 11004, USA; 10Department of Psychiatry and Molecular Medicine, Donald and Barbara Zucker School of Medicine at Hofstra/Northwell, Hempstead, NY 11549, USA; 11Department of Child and Adolescent Psychiatry/Psychotherapy, University Hospital Ulm, 89075 Ulm, Germany; 12Clinical Trial Center Wuerzburg, University Hospital Wuerzburg, 97080 Wuerzburg, Germany; 13Max-Planck-Institut für Psychiatry, 80804 Munich, Germany; 14Specialist Practice and Medical Care Centre for Child and Adolescent Psychiatry Munich, Dr. Epple & Dr. Reuter-Dang, 81241 Munich, Germany; 15Specialist Practice and Medical Care Center for Child and Adolescent Psychiatry Munich, 81241 Munich, Germany; 16Central Information Office, Department of Neurology, Philipps University of Marburg, 35112 Marburg, Germany; 17Parkland-Clinic, Clinic for Psychosomatics and Psychotherapy, Academic Teaching Hospital for the University Gießen, 34537 Bad Wildungen, Germany; 18Institute of Clinical Epidemiology and Biometry, University of Wuerzburg, 97080 Wuerzburg, Germany; 19Department of Psychiatry, Psychosomatics and Psychotherapy, Center of Mental Health, University Hospital of Wuerzburg, 97080 Wuerzburg, Germany; 20Department of Child and Adolescent Psychiatry, Psychosomatics and Psychotherapy, Leopoldina Hospital, 97422 Schweinfurt, Germany; 21Department of Child and Adolescent Psychiatry and Psychotherapy, Central Institute of Mental Health, Medical Faculty Mannheim, Heidelberg University, 68159 Mannheim, Germany; 22Department of Child and Adolescent Psychiatry, Psychotherapy and Psychosomatic medicine, Vivantes Clinic Berlin Neukölln, 12351 Berlin, Germany; 23Clinic for Child and Adolescent Psychiatry, Center for Psychosocial Medicine, University Hospital Heidelberg, 69115 Heidelberg, Germany; 24University Hospital of Child and Adolescent Psychiatry and Psychotherapy, University of Bern, 3000 Bern, Switzerland; 25Department of Child and Adolescent Psychiatry and Psychotherapy, Brandenburg Medical School Brandenburg, 16816 Neuruppin, Germany; 26Department of Child and Adolescent Psychiatry, Neurology, Psychosomatics, and Psychotherapy, University Medical Center Rostock, 18147 Rostock, Germany; 27Department of Child and Adolescent Psychiatry, Psychosomatics and Psychotherapy, University Hospital of Psychiatry and Psychotherapy Tuebingen, Center of Mental Health Tuebingen, 72076 Tuebingen , Germany; 28Clinic for Child and Adolescent Psychiatry, Psychosomatics and Psychotherapy, University of Regensburg at the Regensburg District Hospital, Medbo KU, 93053 Regensburg, Germany; 29Zurich Center for Integrative Human Physiology, University of Zurich, 8057 Zürich, Switzerland; 30Neuroscience Center Zurich, University of Zurich and ETH, 8057 Zürich, Switzerland

**Keywords:** TDM, adolescents, depression, antidepressants, selective serotonin reuptake inhibitors, pharmacovigilance, steady-state concentration, reference range

## Abstract

Fluoxetine is the recommended first-line antidepressant in many therapeutic guidelines for children and adolescents. However, little is known about the relationships between drug dose and serum level as well as the therapeutic serum reference range in this age group. Within a large naturalistic observational prospective multicenter clinical trial (“TDM-VIGIL”), a transdiagnostic sample of children and adolescents (*n* = 138; mean age, 15; range, 7–18 years; 24.6% males) was treated with fluoxetine (10–40 mg/day). Analyses of both the last timepoint and all timepoints (*n* = 292 observations), utilizing (multiple) linear regressions, linear mixed-effect models, and cumulative link (mixed) models, were used to test the associations between dose, serum concentration, outcome, and potential predictors. The receiver operating curve and first to third interquartile methods, respectively, were used to examine concentration cutoff and reference values for responders. A strong positive relationship was found between dose and serum concentration of fluoxetine and its metabolite. Higher body weight was associated with lower serum concentrations, and female sex was associated with lower therapeutic response. The preliminary reference ranges for the active moiety (fluoxetine+norfluoxetine) were 208–328 ng/mL (transdiagnostically) and 201.5–306 ng/mL (depression). Most patients showed marked (45.6%) or minimal (43.5%) improvements and reported no adverse effects (64.9%). This study demonstrated a clear linear dose–serum level relationship for fluoxetine in youth, with the identified reference range being within that established for adults.

## 1. Introduction

Fluoxetine (FLX) was first characterized in a scientific journal in 1974 as a representative of a new class of antidepressant drugs [1,2]. In 1987, FLX was approved by the US Food and Drug Administration (FDA) as the first selective serotonin (5-hydroxytryptamine) reuptake inhibitor (SSRI) for the treatment of depression in adults [3]. It belongs to the most widely used class of antidepressants worldwide and is the most researched antidepressant in childhood and adolescence [4,5,6,7]. FLX is approved by the US FDA in treatment of minors for major depression (>8 years) and obsessive-compulsive disorder (OCD) (>7 years) and by the European Medicines Agency for treatment of minors aged 8 years or more who are suffering from moderate or severe depression [8,9]. A recent umbrella review shows that FLX is the only psychotropic drug studied that is superior to placebo in terms of primary outcome efficacy as well as response and remission in the treatment of depression in children and adolescents [10]. Hetrick et al. (2021) present a somewhat more reserved conclusion in a Cochrane Review and network meta-analysis, asserting that moderately strong evidence indicates FLX can be considered, along with other specific serotonin reuptake inhibitors (SSRIs), as the first choice for treatment of childhood and adolescent depression [4]. Additionally, evidence supports the use of FLX in children and adolescents, usually in combination with cognitive-behavioral therapy (CBT), for anxiety disorders and OCD [10]. Although the evidence is scarce for the effective use of FLX in treatment of bulimia nervosa and binge eating disorder in adolescents, off-label use is occurring [11,12].

FLX is a highly specific serotonin reuptake inhibitor with no or very low affinity to α- and β-adrenoceptors, histamine, and muscarinic receptors and, hence, fewer adverse effects than tricyclic antidepressants and especially fewer cardiovascular and anticholinergic side effects [13]. Although FLX is almost completely absorbed after oral administration, hepatic first-pass metabolism reduces its bioavailability to less than 90%. This medication has the largest volume of distribution of all SSRIs, but the relative concentration in the brain is lower compared to other SSRIs [14]. FLX also has a high protein binding rate of 94.5% [9]. 

Norfluoxetine (NORFLX) is the primary active metabolite and occurs after *N*-demethylation in first-pass hepatic metabolism, mainly by CYP2D6 [5]. Other isoenzymes, like CYP2C9 and CYP2C19, are involved to a lesser extent [5,13,14]. FLX is a potent inhibitor of CYP2D6 and thus inhibits its own metabolism. This can cause nonlinear kinetics with disproportionate increases in serum levels after dose escalation and is responsible for the long half-lives of FLX (4–6 days) and NORFLX (4–15 days) [15]. Additionally, the high binding affinity of FLX and NORFLX to CYP2D6 also causes numerous drug–drug interactions [5]. Under steady-state conditions, the serum levels of NORFLX are normally higher than those of the parent substance FLX [14]. Owing to the similar properties of the parent compound and its metabolite NORFLX, the sum of FLX and NORFLX serum concentrations as the “active moiety” is particularly relevant for therapeutic drug monitoring (TDM) [15].

A study by Meyer et al. (2004) suggests that, for SSRIs, 80% occupancy of serotonin transporters (5-HTT) is necessary for an expected treatment effect in depressive episodes in adults (aged 20–50 years). For FLX, this was achieved on average at a dose of 20 mg in adults [16]. The guideline-recommended and clinically standard dosage for FLX in children and adolescents with depressive disorders aged 8 years and older is a starting dose of 10 mg and a median dose of 20 mg [17,18]. Higher doses, usually up to 60 mg, are prescribed off-label, mainly for the treatment of other disorders such as bulimia nervosa and OCD [19].

About 40–60% of patients treated with FLX do not respond to pharmacologic monotherapy [20,21]. There are many possible influences moderating the effect of treatment with FLX. This is primarily due to pharmacokinetic and pharmacodynamic reasons. Polymorphisms of hepatic enzymes constitute an important class of pharmacokinetic factors that influence serum levels directly. For example, polymorphisms in CYP2D6 lead to different serum concentrations in adults and minors (though CYP2C9 and CYP2C19 polymorphisms seem to have little effect here) [22]. In European/Caucasian populations, 5–10% are poor metabolizers, and about 3% are ultrarapid metabolizers owing to their *CYP2D6* genotype [23,24]. This can lead to significant interindividual variability and have an impact on the desired effect as well as possible adverse effects, even toxic effects. Regarding the toxicity of FLX, case reports of intoxications demonstrate relatively low toxicity compared to tricyclic antidepressants. Postmortem studies and case reports indicate values of 1300–7000 ng/mL for FLX serum levels and 400–4000 ng/mL for NORFLX in fatal overdoses [25,26]. For adults, a laboratory alert value of 1000 ng/mL for the active moiety is reported, which is twice the upper limit of the therapeutic reference range [15]. Overdosing may lead to adverse effects such as tachycardia or drowsiness. This can in some cases already occur at approximately 400 ng/mL for the active moiety with significant interindividual variation [27,28,29]. In minors it appears that the dosage per kg body weight is the most relevant factor. Females and adolescents (compared to males and adults, respectively), seem to be at higher risk for adverse effects of antidepressants, including SSRIs [30]. 

The goal in the treatment of patients is achieving the best possible risk–benefit balance. TDM makes a significant contribution to this objective, as it aims to decrease the frequency and duration of psychiatric episodes by improving the probability of the therapeutic effect through verifying serum levels are within the reference range [15]. Measurement of serum concentrations of the parent compound and its relevant metabolites at steady state allows dosage adjustments within a defined reference range for the best efficacy and lowest possible adverse effect risk. In children and adolescents, developmental pharmacokinetic and pharmacodynamic differences in drug metabolism can play a critical role, but TDM studies in minors are limited [31,32].

To date, only three studies have examined the dosage, serum concentration, and efficacy of FLX in children and adolescents [33,34,35]. There are divergent results on the relationship between dosage and serum concentration of NORFLX and the active moiety as well as on the relationship between serum levels and both therapeutic responses and adverse effects. However, one critical commonality of these studies is their relatively small sample sizes (*n* = 64 to 74).

Therefore, the present naturalistic multicentric study was undertaken to investigate the relationships between dosage, serum concentration, and its predictors in children and adolescents treated with FLX. Additionally, our goal was to identify a preliminary age- and indication-specific therapeutic range of the active moiety, for which we explored serum level associations with clinical effects and adverse effects.

## 2. Materials and Methods

### 2.1. Study Design and Population

This study is part of the “TDM-VIGIL” project, a large-scale, prospective, multicenter pharmacovigilance study conducted by the TDM-VIGIL consortium in partnership with child and adolescent psychiatric centers in Germany, Austria, and Switzerland [36]. The project was funded by the German Federal Institute of Drugs and Medical Devices (BfArM) and received approval from the ethics committee at the primary study center, University Hospital Wuerzburg (301/13), as well as from the local ethics committees of participating centers. This study was conducted in accordance with the Declaration of Helsinki and registered in the European Clinical Trials Database (EudraCT, 2013-004881-33). As the subjects were minors, the parents or guardians had to provide consent for their child to participate in the study. From the age of 14 onward, informed assent was also required from the patients themselves.

The study spanned three different routine healthcare settings, including inpatient, outpatient, and day-treatment units. Patients were included across diagnoses if they started treatment with FLX or if an existing treatment was switched to FLX. All dosing steps were recorded along with corresponding dates in the medication protocol. The steady state (after at least 3.3 half-lives in relation to the lower range limit) was confirmed by the treating physicians and documented in the patient’s medical records. The following assessment time points were incorporated into the study design: baseline (before starting FLX), after reaching the target dose and remaining at a steady state, at discharge from the treatment setting (or in outpatients, when the time interval until the next scheduled follow-up appointment was expected to be 6 months), a follow-up two weeks later, and a follow-up six months after the last steady-state assessment. Depending on the clinical course, additional visit times were possible. The actual timeline and number of visits varied among individual patients.

### 2.2. Patient Assessment

Experienced child and adolescent psychiatrists monitored the treatment course. They diagnosed patients according to the International Classification of Diseases, 10th Revision (ICD-10). Indication for FLX was assessed and recorded. A medical examination for each patient was conducted, which included checking vital signs, electrocardiography, and laboratory blood tests to assess hepatic and renal function.

On the day of blood collection, treatment responses were measured using the Clinical Global Impression (CGI) Scale (Improvement, CGI-I), which clinicians were instructed to rate solely based on the effects of the drug. The scale scores for global improvement were as follows: ‘very much improved’, ‘much improved’, ‘minimally improved’, ‘no change’, ‘minimally worse’, ‘much worse’, ‘very much worse’. To measure adverse reactions to FLX, the Pediatric Adverse Event Rating Scale (PAERS) was employed [37], as it is considered to be a valid and informative assessment tool for pharmacotherapy studies [15]. This instrument was specifically designed to quantify the severity of 56 signs of adverse events occurring in pediatric patients under treatment with a psychotropic drug. The 56 items were rated on a Likert scale with ‘none’, ‘slight’, ‘moderate’, ‘severe’, and ‘extremely severe’ responses. The maximal severity on any of the items was considered in this study. Only adverse effects related to FLX, and not to concurrent psychiatric or somatic medications, were included as relevant for this study. Additionally, indications for the TDM measurement, detailed information on concomitant medications, nicotine use, weight, and height were also recorded. Non-compliance was rated using a pre-defined rating schema (Appendix A), which was documented at the time of blood collection/serum concentration measurement.

### 2.3. Serum Concentration Analysis

TDM analysis was conducted according to the guidelines of the Working Group on Neuropsychopharmacology and Pharmacopsychiatry (AGNP) consensus [15]. Blood samples were collected from cubital veins into 7.5-mL monovettes without any additives or anticoagulants at a steady-state trough level. The samples were immediately analyzed (in the case of samples from Wuerzburg, Germany) or analyzed after being received by the TDM laboratory in Wuerzburg, Germany, shipped by regular mail. The concentrations of FLX and NORFLX were determined using an isocratic reversed-phase high-performance liquid chromatography method with UV-absorbance detection on an Agilent 1200 series system (Agilent Technologies Inc., Santa Clara, CA, USA), as described in detail in the Appendix A. Each analytical series included internal quality control samples, and external control samples were analyzed quarterly. The laboratory responsible for the analysis is certified by a quality control program (https://www.instand-ev.de, accessed on 2 July 2023; https://www.ukw.de/psychiatrie/zuweisende-kolleginnen-und-kollegen/tdm-labor/zertifikate/, accessed on 2 July 2023). Analytical grade chemicals and solvents were procured from Sigma-Aldrich Chemie GmbH (Taufkirchen, Germany). 

For further information on the study design, see Egberts et al. (2022) [38].

### 2.4. Data Management and Statistical Analysis

All statistical analyses were performed using R (R Foundation for Statistical Computing) version 4.1.1. The alpha level for all analyses was set to 0.05. To maximize the comparability of the conclusions from the dataset generated in this study, the main analyses were performed with both the last valid timepoint, as in previous relevant studies [33,39], and with all the valid timepoints, as detailed in the next paragraph. Each analysis block was conducted separately for FLX, its active metabolite NORFLX, and the active moiety (FLX+NORFLX), which exhibited anticipated correlations with each other. First, the relationship between the daily dose of FLX (in mg) and serum concentrations (in ng/mL) were examined using linear regressions (the *lm* function). The model assumptions were verified by means of the semi-automated diagnostic functions in the ‘gvlma’ (Global Validation of Linear Models Assumptions) R package, including evaluation of global statistics, skewness, kurtosis, link function, and heteroscedasticity parameters and inspection of the related plots [40]. As not all the assumptions were met by the original data, a square-root transformation was applied to the outcome variables. In addition to the simple linear regressions, multiple linear regressions were performed with four additional hypothesized predictors: age (years), sex (male/female), body weight (kg), and smoking status (yes/no). The variance inflation factor (VIF) was evaluated as a measure of multicollinearity. Both regressions were compared based upon model fit. Second, proportional odds models using the *clm* (cumulative link models) function in the ‘ordinal’ R package were fitted to examine the serum concentrations as predictors of clinical effects (measured by CGI-I) and adverse effects (measured by PAERS**—**maximal severity). As there was only a single ‘very severe’ PAERS response, ‘very severe’ and ‘severe’ responses were pooled together. The analyses were repeated with an extended set of hypothesized predictors. For CGI-I, these were age, sex, psychotropic co-medication (i.e., antipsychotic and antidepressant drug), and diagnostic comorbidity (i.e., a single diagnosis versus multiple diagnoses). For PAERS, the predictors were age, sex, and psychotropic co-medication. The proportional odds assumptions in these statistics were verified with likelihood ratio tests using the *nominal_test* and *scale_test* functions, while model convergence was tested with the *convergence* function. 

In the analyses of all datapoints, the *lmer* (linear mixed-effect model) function from the ‘lme4’ package was used to investigate the associations with daily dose of FLX and serum concentrations using square-root-transformed FLX, NORFLX, and FLX+NORFLX concentrations (separately) as outcome variables, dose as a fixed predictor, and subject ID as a random intercept. The *clmm* (cumulative link mixed models) function from the ‘ordinal’ package was used to test the impact of metabolite level on clinical and adverse effects with CGI-I and PAEARS (separately) as outcome variables, FLX, NORFLX, and FLX+NORFLX concentrations (separately) as fixed predictors, and subject ID as a random intercept. The model with multiple predictors included the variables, as in the single timepoint analyses, as well as the data collection time (i.e., days from the baseline timepoint).

To determine a possible concentration threshold of efficacy, receiver operating curve (ROC) analyses were performed based on a binary logistic regression, and the optimal cutoff was calculated by finding an optimal trade-off between sensitivity and specificity. The “good responders” and “poor/non-responders” groups were constructed by pooling the CGI-I responses of ‘very much improved’ and ‘much improved’ as “good responders” and ‘minimally improved’, ‘no change’, ‘minimally worse’, ‘much worse’, and ‘very much worse’ as “poor/non-responders”. Finally, we determined an effective concentration level range (i.e., reported for good clinical responders to FLX) as the mean ± one standard deviation and the 25th–75th interquartile range, which approximate the reference therapeutic range according to previous recommendations [41]. The last valid datapoints were included in these analyses as deemed most appropriate. Transdiagnostic responders and responders with depression (F32.X and F33.X) were considered separately.

The quality control criteria for data exclusion were the interval between blood collection and laboratory analysis exceeding 72 h, uncertain patient medication compliance noted by a clinician, and blood not being collected during steady-state conditions. In addition, anorexia nervosa patients with a body mass index (BMI) lower than the 10th percentile were also excluded. 

## 3. Results

### 3.1. Descriptive Statistics

Study population characteristics expressed as means with standard deviations or counts with corresponding percentages are listed in Table 1. After excluding patients with data that did not meet the quality control criteria and/or missing data (missing dose, blood, or CGI/PAERS information), 138 patients were included in this study (mean age, 15 years; age range, 7–18; 34 males [24.6%]). The mean daily FLX dosage was 19.93 ± 5.30 mg (range, 10–40 mg), and the related mean steady-state serum concentration for FLX was 123.66 ± 65.17 ng/mL (range, 17–396 ng/mL), for NORFLX was 144.74 ± 65.93 ng/mL (range, 21–339, ng/mL), and for FLX+NORFLX was 264.21 ± 110.85 ng/mL (range, 62–673 ng/mL). Most patients responded with marked improvement (38.4% ‘much improved’ and 7.2% ‘very much improved’), followed by minimal improvement (43.5%). The majority reported no adverse effects (64.9%), followed by moderate (19.4%) and then slight (13.4%) adverse effects. 

The number of timepoints in the analyses including more than one observation ranged from two to four. There were between 292 and 281 observations in each of the analyses involving dosage and serum concentrations and between 216 and 205 observations in each of the analyses involving serum concentrations and clinical/adverse effects. 

### 3.2. Dose and Metabolite Concentrations (Last Datapoints)

In the simple linear regressions (FLX, *F*_1,136_ = 16.84, *p* < 0.001, adj. *R*^2^ = 0.104; NORFLX, *F*_1,132_ = 19.07, *p* < 0.001, adj. *R*^2^ = 0.120; FLX+NORFLX, *F*_1,136_ = 22.34, *p* < 0.001, adj. *R*^2^ = 0.135) there was a positive association between dosage and serum concentrations (FLX, *β* = 0.18, CI 0.09–0.26, *p* < 0.001; NORFLX, *β* = 0.19, CI 0.10–0.27, *p* < 0.001; FLX+NORFLX, *β* = 0.24, CI 0.14–0.34, *p* < 0.001). Scatter plots showing these effects for the untransformed outcome variables are presented in Figure 1. In the multiple linear regressions, two predictors, dose and body weight, were significant (Table 2A). All VIF values were lower than 2, indicating there were no multicollinearity issues. All models with multiple predictors explained significantly more variance than the simple models (FLX, adj. *R*^2^_M2−M1_ = 0.153, *SS* = 188.0, *F*_4,132_ = 8.003, *p* < 0.001; NORFLX, adj. *R*^2^_M2−M1_ = 0.092, *SS* = 116.8, *F*_4,128_ = 4.857, *p* = 0.001; FLX+NORFLX, adj. *R*^2^_M2−M1_ = 0.156, *SS* = 272.46, *F*_4,131_ = 8.679, *p* < 0.001). 

### 3.3. Dose and Metabolite Concentrations (All Datapoints)

Inclusion of all datapoints confirmed the findings from the last datapoint analysis showing a strong positive effect of dose on concentration (FLX, *β* = 0.31, CI 0.25–0.36, *p* < 0.001, marginal *R*^2^ *=* 0.285, conditional *R*^2^ = 0.562; NORFLX, *β* = 0.25, CI 0.20–0.30, *p* < 0.001, marginal *R*^2^ *=* 0.216, conditional *R*^2^ = 0.679; FLX+NORFLX, *β* = 0.38, CI 0.31–0.44, *p* < 0.001, marginal *R*^2^ *=* 0.311, conditional *R*^2^ = 0.548). In all three models with multiple predictors, the effects of dose and body weight were significant (Table 2B). In addition, there was a significant effect of time (positive direction) in the model including FLX and a significant effect of sex (maleness being a negative predictor) in the model including NORFLX.

### 3.4. Metabolite Concentrations and Clinical/Adverse Effects (Last Datapoints)

The simple ordinal regressions with CGI-I responses as outcome variables revealed no statistically significant effect for any of the three serum concentration parameters (all *p* > 0.184). Similarly, there were no significant results for PAERS as a measure of adverse reactions (all *p* > 0.481). Among the covariates, there was a significant effect of sex (being male was a positive predictor of clinical efficacy) in all three models with CGI-I (Table 3A,B).

### 3.5. Metabolite Concentrations and Clinical/Adverse Effects (All Datapoints)

No significant effects were observed in the simple linear models involving CGI-I (all *p* > 0.169) or PAERS (all *p* > 0.469). In all three models including CGI-I with multiple predictors, there was a significant positive effect of maleness (Table 3C). No statistically significant effect of predictors were found in the models including PAERS (Table 3D).

To additionally explore the role of diagnosis, we entered five diagnostic categories as a predictor (i.e., affective disorders, anxiety disorders, obsessive-compulsive disorders, eating disorders, other disorders) in each of the above models. All the described effects remained significant with ‘other disorders’ being the only additionally significant diagnosis with reference to the affective disorder group (FLX: OR = 11.84, CI = 3.12–44.94, *p* < 0.001; NORFLX: OR = 10.57, CI = 2.97–37.63 *p* < 0.001; FLX+NORFLX: OR = 11.78, CI = 3.10–44.72, *p* < 0.001) in the model with PAERS with all datapoints. The same model with PAERS with the last point did not converge. 

### 3.6. ROC Analysis and Effective Interquartile Ranges

The ROC analysis indicated unsatisfactory discrimination between poor (*n* = 74) and good responders (*n* = 64) transdiagnostically (FLX, area under the curve [AUC] = 0.491 [95% CI, 0.392–0.59]; NORFLX, AUC = 0.437 [95% CI, 0.338–0.535]; FLX+NORFLX, AUC = 0.427 [95% CI, 0.33–0.524]). Similar results were found between poor (*n* = 47) and good responders (*n* = 44) with a main diagnosis of depression (FLX, AUC = 0.496 [95% CI, 0.376–0.617]; NORFLX, AUC = 0.469 [95% CI, 0.347–0.591]; FLX+NORFLX, AUC = 0.451 [95% CI, 0.332–0.571]). The mean ± 1 standard deviation and 25th–75th interquartile concentration ranges for FLX, NORFLX, and FLX+NORFLX are listed in Table 4.

## 4. Discussion

The aim of this study was to examine the relationships between dosage, serum concentration, and its predictors in children and adolescents treated with FLX. We also explored serum level associations with clinical/adverse effects and determined a tentative age-/indication-specific therapeutic range of the active moiety. The results indicated a significant association between the FLX dose and the serum levels of FLX, NORFLX, and the active moiety, which was observed for the last timepoint and was even more pronounced when considering all serum level datapoints collected. Body weight was significantly and inversely associated with the serum concentrations in both models (holding other predictors constant). In terms of efficacy, there was no significant relationship between serum levels of FLX, NORFLX, or the active moiety and therapeutic effect as assessed by CGI-I. Among the variables considered, females showed lower odds of clinical effect (holding other predictors constant). In terms of adverse effects measured by PAERS, there was no significant association with serum levels, and no other tested variable was found to be a significant predictor.

The observed associations between daily FLX dose and serum concentrations of FLX, NORFLX, and the active moiety together with the impact of body weight are in accordance with a study by Blázquez et al. (2014) [34] in adolescents and with a study by Lundmark et al. (2001) in adults [42]. Koelch et al. (2012) observed a positive correlation between the dose of FLX per kg of body weight and serum levels of FLX but not for NORFLX and the active moiety [33]. Other study results in adults revealed a similar but age-dependent heterogeneous pattern [43]. A possible explanation for these inconsistencies could be phenoconversion. Specifically, drug-induced altered enzyme activity modifies the genetically determined metabolism. The inhibition of CYP2D6 enzyme activity by FLX impairs its own metabolism and may lead to an increased FLX-to-NORFLX ratio [43,44,45]. In about 20–30% of patients taking an enzyme inhibitor of CYP2D6, such as FLX, this leads to phenoconversion to poor metabolizer status [46]. Influencing factors appear to include genetic vulnerability, dosage, and length of drug exposure [44]. While dosage and duration of exposure were considered in most statistical analyses of the mentioned studies, no conclusion can be drawn regarding genetic vulnerability. 

Koelch et al. observed a lower serum level for the active moiety and NORFLX in smokers [31]. This was not replicated in our data, but it is worth noting that the proportion of smokers was considerably lower in our study (5.7% regular smokers versus 20% in the study by Koelch and colleagues), likely obscuring such a possible effect. Notably, a study of adults including a large proportion of smokers (24%) also found no difference regarding the active moiety or NORFLX serum levels [42]. Blázquez et al. did not include smoking in their analyses [32]. Nicotine is not a known general inducer or inhibitor of CYP2D6 and thus is not expected to affect serum concentrations of FLX through known mechanisms [47]. Only in the brain does nicotine appear to induce CYP2D6 expression [48]. 

Consistent with other TDM studies on FLX conducted in this age group, we did not find any age effect. Many factors determining pharmacokinetics mature within the first two years of life [49]. Interindividual genetic variation in CYP2D6 activity also seems to have more influence on its activity than developmental aspects in late childhood and adolescence [50]. This may explain why no effect of age was found in the present study population of individuals aged 7–18 years, which comprised predominantly adolescents. 

In our data, males tended to show lower levels of NORFLX, which may be related to the higher average CYP2D6 activity in females [51]. Koelch et al. found no evidence that sex exerts a moderating influence on serum concentration of FLX and its metabolite NORFLX [31]. However, Blázquez et al. in minors and Amsterdam et al. in adults found higher FLX, NORFLX, and the active moiety concentrations in females [34,52]. There is convincing evidence of pharmacokinetic differences between the sexes, for example, related to the volume of distribution and drug clearance. This often leads to higher serum concentrations in females [51,53]. The heterogeneous findings are probably owing to the numerous influencing factors, such as genetic polymorphism in *CYP2D6* and interindividual pharmacokinetic differences, which were not controlled in these studies. 

The reference range calculated in our study for responders with depression was 201.5–306 ng/mL, which is within the reference range for adults (120–500 ng/mL) [15]. When all diagnostic groups were considered, the range was slightly higher (208–328 ng/mL). 

In line with other comparable studies, there was no association of therapeutic response with serum levels of FLX, NORFLX, or the active moiety in our study. The lack of an association between FLX treatment plasma levels and therapeutic response has been repeatedly observed both in adults and children and adolescents [33,34,52]. A recent meta-analysis also suggests that no such association exists for other SSRIs. In addition, for adults, high doses of FLX (60 mg/day) were shown to be less effective than 20 mg/day [54]. A key exception is reported by Sakolsky et al. (2011), who observed a non-significant trend in adolescents (12–18 years) for a higher likelihood of response to FLX when patients had serum concentrations above the geometric mean compared with those who were below it, as well as a significantly higher likelihood of response when the dose was increased, thus raising serum concentrations from below the geometric mean to above it [35]. Small samples, a lack of systematic and detailed symptom assessments, and several other confounders may explain the failure to detect an association between dose or serum level and response in the vast majority of studies. 

First, genetic variants may influence the efficacy of SSRIs in treating major depression and other psychiatric disorders, as exemplified by a polymorphism of the *SCL6A4* gene that encodes the sodium-dependent serotonin transporter (5-hydroythyptamine transporter). Although study results are heterogeneous, there is evidence that different polymorphisms are associated with response to FLX and other SSRIs or treatment failure [55,56]. Studies of anxiety disorders and obsessive-compulsive disorders in addition to depression have also shown that other genetic variants in genes encoding both enzymes involved in serotonin biosynthesis (tryptophan hydroxylase, TPH2) and serotonin receptors (5-hydroxytryptamine-receptor, HTR1B) also have an impact on clinical outcomes in children and adolescents [57]. For the *ABCB1* gene, which encodes P-glycoprotein, a protein that transports harmful substances out of the cell to protect it, there is also evidence that polymorphisms influence the therapeutic effect of FLX in children and adolescents [22,58].

Second, in study populations with depressive disorders in particular, placebo responders, verum responders, and non-responders tend to each comprise a third of all subjects. This results in a low signal-to-noise ratio and blurs the true concentration–response association [41]. A meta-analysis showed that to detect the effect of SSRIs administered in childhood and adolescence, the number of subjects needed to treat in randomized controlled trials is nine on average and higher in younger patients [59].

Third, as argued by Meyer et al. (2004) based on positron emission tomography imaging studies, observed saturation dynamics of inhibition of the serotonin transporter suggest a nonlinear concentration–response relationship rather than a linear relationship [16]. However, this is not supported by our results. Our ROC analysis showed no evidence of a FLX concentration threshold differentiating good responders from poor responders. Nevertheless, 80% saturation is already achieved at a dose of 20 mg, which was usually the targeted dose in our study.

Notably, our results also indicated higher odds of males (versus females) benefiting from FLX in terms of clinical improvement, which contrasts with the available literature [60,61]. Although there are several studies that suggest males are more likely to benefit from tricyclic antidepressants than females, for SSRIs, either no difference or even a better response in females, especially among those of reproductive age, was found [61,62,63]. Therefore, our finding that being male is a significant predictor of clinical response to SSRIs is not easily interpretable. One consideration was that the significant proportion of female patients with anorexia (*n* = 13) could potentially may have had an influence here, even though we excluded underweight female patients. However, the exploratory analysis with diagnostic groups as predictors in the models did not yield a significant result. Pharmacokinetic differences have been suggested as a mechanism causing sex differences in efficacy in general [61]. However, we observed a trend-level effect for a smaller increase in serum levels in males compared to females only for NORFLX, but this would not explain the observed superior effectiveness. 

Overall, there is no conclusive prior data analysis demonstrating a difference between sexes in response to SSRIs. Presumably, numerous possible confounders play some role in our results (e.g., disease severity, adherence, and pharmacokinetic as well as pharmacodynamic sex differences) [61]. 

Like Koelch et al. and Blázquez et al., we could not find an association between serum levels and adverse effects [33,34]. In general, there is compelling evidence that FLX administered for the treatment of depression is well tolerated compared with other antidepressants in childhood and adolescence [7,10]. In our study, the majority of participants reported no adverse effects (64.9%).

The strengths of this study include its extensive analytic strategy maximizing the use of data as well as its large sample size compared to similar prior studies. Participants were recruited from everyday clinical practice through a multicenter approach, making the patient sample representative of real-world clinical care. The study adhered to quality criteria for TDM studies, including using serum analyses and collecting blood samples under steady-state conditions to establish a therapeutic reference range. This was calculated for all diagnoses and separately for the approved indication of depression. In addition, conservative exclusion criteria were applied to avoid data bias, e.g., exclusion of underweight patients.

The results of this study should be considered through the lens of its limitations. First, this is a naturalistic study conducted under some uncontrolled conditions, with a flexible dosing strategy and the inclusion of patients with concomitant psychotropic and somatic medications. While psychotropic co-medication was taken into account for efficacy assessment, influences on the therapeutic response that depend on the target symptomatology cannot be excluded. Second, we did not consider co-medication for the dose-serum concentration correlations. An effect here would have been expected only from drugs known to interact with CYP2D6, and the group of patients with corresponding co-medication was too small (*n* = 2) to influence the findings substantially. Third, participants were included for whom prior treatment with another psychotropic drug was switched to FLX owing to a lack of response. This resulted in a heterogeneous group with varying degrees of clinical severity and response status, which complicated the identification of a possible relationship between medication and therapeutic response, which may have additionally been hampered by the use of a broad transdiagnostic instrument like the CGI-I. Fourth, in this naturalistic observational study, patients from all diagnostic groups were included. Although the identified reference range for depression did not significantly differ from the transdiagnostic range, due to the typically distinct neurobiological etiology of each disorder, only diagnosis-specific statements can be made. Fifth, enantiomers were not differentiated in the study, despite the considerable interindividual variation revealed by previous research [64]. Sixth, due to clinical practicability, the prompt detection of potential poor metabolizers, and the capture of serum levels for early treatment effects within the first two weeks, we opted for the earliest possible time point as steady state [65]. It is important to note that this choice could potentially result in reference ranges that are rather underestimated. Seventh, statistical modeling of multi-level data from a naturalistic study, which prioritizes the clinical objectives and leaves some factors less controllable or even unmeasured, can be a challenging task. In general, the goodness-of-fit indices for our more complex models were relatively low. Future studies should explore additional explanatory variables and employ different data collection strategies. Finally, there was no rigorous control of medication adherence, although practitioner assessments were collected for each patient to mitigate this issue.

## 5. Conclusions

This study reinforces recent evidence of a linear relationship between FLX dosing and serum concentrations of FLX, NORFLX, and the active moiety in children and adolescents. It also elucidates some other clinical variables, which may be of relevance in this age group. Given the inverse association between weight and serum concentrations, in further studies over- or underweight status should be considered as an influencing factor that may require dose adjustment. The lack of significant associations between serum concentration and clinical/adverse effects, as shown in numerous naturalistic studies, may be owing to uncontrolled confounders and a low signal-to-noise ratio, as well as complex interactions and genetic polymorphisms that influence pharmacodynamics. More sophisticated study designs are needed to isolate possible differential contributions to clinical response to antidepressants in the future.

## Figures and Tables

**Figure 1 pharmaceutics-15-02202-f001:**
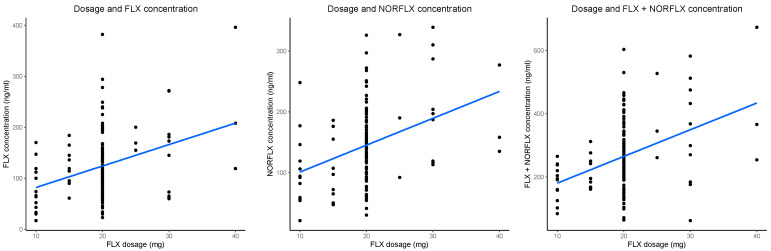
Significant linear associations between fluoxetine (FLX) daily dose (*x*-axis) and steady-state serum concentrations of FLX, norfluoxetine (NORFLX), and the active moiety (FLX+NORFLX) (*y*-axis; from left to right). The blue diagonal line is the best-fit line. The black dots represent data points.

**Table 1 pharmaceutics-15-02202-t001:** Demographic and clinical characteristics of the study sample.

Characteristic	Overall (*n* = 138)
**Sex**	
male	34 (24.6%)
female	104 (75.4%)
**Age** *	
Mean (SD)	15
Range	7–18
**Age group**	
children (7–12 years)	7 (5.1%)
adolescents (13–18 years)	131 (94.9%)
**Medication dosage**	
Mean (SD)	19.93 (5.30)
Range	10.00–40.00
**FLX concentration**	
Mean (SD)	123 (65.17)
Range	17–396
**NORFLX concentration**	
N-Miss	4
Mean (SD)	144 (65.93)
Range	21–339
**FLX+NORFLX concentration**	
Mean (SD)	264 (110.85)
Range	62–673
**Body weight**	
Mean (SD)	59.67 (16.99)
Range	24.40–124.70
**Body mass index**	
Mean (SD)	21.81 (4.94)
Range	13.88–43.66
**Smoking**	
11–20 cigarettes/day	2 (1.4%)
6–10 cigarettes/day	1 (0.7%)
Up to 5 cigarettes/day	5 (3.6%)
Occasionally	9 (6.5%)
No	121 (87.7%)
**Antipsychotic co-medication**	
No	128 (92.8%)
Yes	10 (7.2%)
**Other antidepressant co-medication**	
No	134 (97.1%)
Yes	4 (2.9%)
**Treatment modality**	
outpatient	8 (5.8%)
inpatient	101 (73.2%)
day care treatment	29 (21.0%)
**Clinical Global Impression—Severity**	
normal, not at all ill	2 (1.4%)
borderline mentally ill	4 (2.9%)
mildly ill	21 (15.2%)
moderately ill	61 (44.2%)
markedly ill	37 (26.8%)
severely ill	13 (9.4%)
among the most extremely ill patients	0 (0%)
**Clinical Global Impression—Improvement**	
no assessment possible	2 (1.4%)
very much improved	10 (7.2%)
much improved	53 (38.4%)
minimally improved	60 (43.5%)
no change	8 (5.8%)
minimally worse	3 (2.2%)
much worse	2 (1.4%)
very much worse	0 (0%)
**Pediatric Adverse Events Rating Scale (Max. Severity)**	
N-Miss	4
none	87 (64.9%)
slight	18 (13.4%)
moderate	26 (19.4%)
severe	2 (1.5%)
extremely severe	1 (0.7%)
**Main diagnosis (ICD-10)** Major depressive disorders (MDD)	
F32.0 MDD, mild episode	2 (1.4%)
F32.1 MDD, moderate episode	63 (45.7%)
F32.2 MDD, severe episode	19 (13.8%)
F32.3 MDD, severe episode with psychotic symptoms	2 (1.4%)
F33.1 recurrent MDD, moderate episode	6 (4.3%)
F33.2 recurrent MDD, severe episode	1 (0.7%)
Anxiety disorders	
F40.1 social phobias	5 (3.6%)
F40.2 specific phobias	1 (0.7%)
F41.2 mixed anxiety and depressive disorder Obsessive-compulsive disorders	3 (2.2%)
F42.2 mixed obsessional thoughts and acts Reaction to severe stress, and adjustment disorders	1 (0.7%)
F43.1 posttraumatic stress disorder (PTSD)	1 (0.7%)
F43.2 adjustment disorders Somatoform disorders	1 (0.7%)
F45.1 undifferentiated somatoform disorder Eating disorders	1 (0.7%)
F50.0 anorexia nervosa	11 (7.9%)
F50.1 atypical anorexia nervosa	2 (1.4%)
F50.2 bulimia nervosa	4 (2.9%)
F50.3 atypical bulimia nervosa Personality disorders	2 (1.4%)
F60.31emotionally unstable, borderline Pervasive developmental disorders	4 (2.9%)
F84.1 atypical autism Behavioral and emotional disorders	1 (0.7%)
F91.- conduct disorder	1 (0.7%)
F92.0 depressive conduct disorder	5 (3.6%)
F94.0 elective mutism	2 (1.4%)
**Diagnostic groups by sex**	
**Affective disorders**	
male	23 (67.6%)
female	70 (67.3%)
**Anxiety disorders**	
male	4 (11.8%)
female	5 (4.8%)
**Obsessive-compulsive disorders**	
male	0
female	1 (1.0%)
**Eating disorders**	
male	0
female	19 (18.3%)
**Other disorders**	
male	7 (20.6%)
female	9 (8.7%)
**Co-morbidity**	
1 diagnosis	70 (50.7%)
≥2 diagnoses	68 (49.3%)

Abbreviations*:* FLX, fluoxetine; FLX+NORFLX, active moiety; ICD-10, International Classification of Diseases (10th edition); N-Miss, missing data; NORFLX, norfluoxetine; SD, standard deviation; * age was entered in the analyses with decimal values.

**Table 2 pharmaceutics-15-02202-t002:** Analyses with serum concentrations and multiple predictors.

A. Model 1 (Last Datapoint)	Multiple Linear Regressions								
	FLX^(1/2)^			NORFLX^(1/2)^			FLX+NORFLX^(1/2)^		
Predictors	Est.	CI	*p*	Est.	CI	*p*	Est.	CI	*p*
(Intercept)	9.99	6.49–13.48	<0.001	8.82	5.19–12.45	<0.001	13.42	9.38–17.46	<0.001
Dose	0.24	0.15–0.32	**<0.001**	0.21	0.13–0.30	**<0.001**	0.30	0.20–0.40	**<0.001**
Age	0.02	−0.23–0.28	0.862	0.12	−0.15–0.38	0.385	0.11	−0.19–0.41	0.474
Sex [male]	−0.08	−1.09–0.92	0.867	−0.97	−2.00–0.06	0.064	−0.79	−1.95–0.37	0.178
Body weight	−0.07	−0.10–−0.04	**<0.001**	−0.05	−0.08–−0.02	**0.001**	−0.08	−0.11–−0.05	**<0.001**
Smoking [yes]	−0.02	−1.81–1.77	0.981	−0.13	−1.94–1.68	0.890	−0.04	−2.10–2.03	0.971
Observations	138			134			138		
*R*^2^/*R*^2^ adjusted	0.284/0.257			0.241/0.212			0.320/0.294		
**B. Model 2** **(All Datapoints)**	**Linear Mixed Effect Models**								
	**FLX^(1/2)^**			**NORFLX^(1/2)^**			**FLX+NORFLX^(1/2)^**		
**Predictors**	**Est.**	**CI**	** *p* **	**Est.**	**CI**	** *p* **	**Est.**	**CI**	** *p* **
(Intercept)	9.40	6.69–12.12	<0.001	8.19	5.23–11.16	<0.001	12.55	9.43–15.68	<0.001
Dose	0.34	0.29–0.40	**<0.001**	0.26	0.20–0.31	**<0.001**	0.41	0.34–0.47	**<0.001**
Age	−0.09	−0.28–0.10	0.369	0.10	−0.10–0.31	0.328	0.02	−0.20–0.24	0.886
Sex [male]	−0.29	−1.06–0.48	0.463	−0.85	−1.70–−0.01	**0.047**	−0.80	−1.69–0.08	0.074
Body weight	−0.07	−0.09–−0.05	**<0.001**	−0.05	−0.08–−0.03	**<0.001**	−0.09	−0.11–−0.06	**<0.001**
Smoking [yes]	−0.02	−1.39–1.36	0.982	0.04	−1.24–1.31	0.953	0.12	−1.46–1.71	0.878
Time	0.004	0.001–0.01	**0.019**	0.00	−0.00–0.00	0.390	0.00	−0.00–0.01	0.061
**Random Effects**									
σ2	4.03			2.68			5.39		
τ_00 ID_	1.72			3.47			2.24		
ICC	0.30			0.56			0.29		
Observations	287			281			287		
*R*^2^/*R*^2^ adjusted	0.407/0.584			0.282/0.687			0.425/0.594		

Abbreviations: FLX, fluoxetine; FLX+NORFLX, active moiety; ICC, interclass correlation coefficient; NORFLX, norfluoxetine; σ^2^, within-person variance; τ_00 ID_, between-person variance.

**Table 3 pharmaceutics-15-02202-t003:** Analyses with serum concentrations and clinical/adverse effects.

**A. CGI-I** **Model 1 (Last Datapoints)**	**Cumulative Link Models (Ordinal Regressions)**								
	**CGI-I (FLX)**			**CGI-I (NORFLX)**			**CGI-I (FLX+NORFLX)**		
**Predictors**	**Est.**	**CI**	** *p* **	**Est.**	**CI**	** *p* **	**Est.**	**CI**	** *p* **
FLX/NORFLX/FLX+NORFLX	1.00	0.99–1.00	0.732	1.00	1.00–1.01	0.089	1.00	1.00–1.01	0.168
Age	1.13	0.94–1.35	0.196	1.10	0.91–1.32	0.334	1.11	0.93–1.33	0.238
Sex [male]	2.66	1.23–5.74	**0.013**	3.05	1.37–6.80	**0.006**	2.89	1.33–6.27	**0.007**
Co-medication [no]	0.52	0.18–1.49	0.226	0.43	0.14–1.30	0.136	0.51	0.18–1.46	0.211
Comorbidity [1 diagn.]	0.79	0.41–1.54	0.493	0.85	0.43–1.68	0.633	0.81	0.41–1.57	0.529
Observations	136			132			136		
*R*^2^ Nagelkerke	0.079			0.208			0.093		
**B. PAERS** **Model 2 (Last Datapoints)**	**Cumulative Link Models (Ordinal Regressions)**								
	**PAERS (FLX)**			**PAERS (NORFLX)**			**PAERS (FLX+NORFLX)**		
**Predictors**	**Est.**	**CI**	** *p* **	**Est.**	**CI**	** *p* **	**Est.**	**CI**	** *p* **
FLX/NORFLX/FLX+NORFLX	1.00	0.99–1.00	0.437	1.00	0.99–1.00	0.753	1.00	1.00–1.00	0.513
Age	0.99	0.82–1.20	0.934	0.98	0.80–1.19	0.837	1.00	0.82–1.21	0.990
Sex [male]	0.92	0.39–2.16	0.851	0.94	0.39–2.23	0.883	0.90	0.38–2.11	0.808
Co-medication [no]	0.51	0.17–1.48	0.214	0.69	0.23–2.08	0.516	0.52	0.18–1.51	0.232
Observations	134			130			134		
*R*^2^ Nagelkerke	0.018			0.095			0.016		
**C. CGI-I** **Model 2 (All Datapoints)**	**Cumulative Link Mixed Models**								
	**CGI-I**			**CGI-I**			**CGI-I**		
**Predictors**	**Est.**	**CI**	** *p* **	**Est.**	**CI**	** *p* **	**Est.**	**CI**	** *p* **
FLX/NORFLX/FLX+NORFLX	1.00	0.99–1.01	0.842	1.01	1.00–1.01	0.160	1.00	1.00–1.01	0.245
Age	1.06	0.81–1.39	0.676	1.03	0.77–1.38	0.845	1.06	0.79–1.42	0.690
Sex [male]	3.55	1.09–11.48	**0.035**	4.06	1.06–15.50	**0.040**	4.20	1.05–16.84	**0.043**
Co-medication [no]	0.76	0.17–3.33	0.719	0.68	0.14–3.27	0.633	0.78	0.16–3.73	0.751
Comorbidity [1 diagn.]	0.44	0.17–1.15	0.093	0.49	0.16–1.48	0.207	0.42	0.13–1.33	0.138
Time	1.00	1.00–1.01	0.086	1.00	1.00–1.01	0.096	1.00	1.00–1.01	0.145
**Random Effects**									
σ2	3.29			3.29		3.29			
τ00 ID	5.07			5.91		6.18			
ICC	0.61			0.64		0.65			
Observations	213			208		213			
Marginal *R*^2^/Conditional *R*^2^	0.066/0.633			0.073/0.669		0.073/0.678			
**D. PAERS** **Model 2 (All Datapoints)**	**Cumulative Link Mixed Models**								
	**PAERS**			**PAERS**			**PAERS**		
**Predictors**	**Est.**	**CI**	** *p* **	**Est.**	**CI**	** *p* **	**Est.**	**CI**	** *p* **
FLX/NORFLX/FLX+NORFLX	1.00	0.99–1.00	0.548	1.00	1.00–1.01	0.570	1.00	1.00–1.00	0.937
Age	1.07	0.87–1.32	0.536	1.05	0.86–1.27	0.658	1.07	0.87–1.32	0.527
Sex [male]	0.95	0.38–2.37	0.911	0.98	0.42–2.31	0.966	0.95	0.38–2.36	0.908
Co-medication [no]	0.62	0.19–1.97	0.414	0.79	0.26–2.37	0.676	0.63	0.20–2.00	0.434
Time	1.00	0.99–1.00	0.536	1.05	0.99–1.00	0.463	1.00	0.99–1.00	0.451
**Random Effects**									
σ2	3.29			3.29		3.29			
τ00 ID	1.36			0.89		1.34			
ICC	0.29			0.21		0.29			
Observations	210			205		210			
Marginal *R*^2^/Conditional *R*^2^	0.015/0.303			0.009/0.220		0.013/0.299			

Abbreviations*:* CGI-I, Clinical Global Impression**—**Improvement; FLX, fluoxetine; FLX+NORFLX, active moiety; ICC, interclass correlation coefficient; NORFLX, norfluoxetine; PAERS, Pediatric Adverse Events Rating Scale. σ^2^, within-person variance; τ_00 ID_, between-person variance.

**Table 4 pharmaceutics-15-02202-t004:** Serum concentrations for responders to fluoxetine.

	Responders Transdiagnostic *n* = 64			Responders with Depression *n* = 44		
	FLX	NORFLX	FLX+NORFLX	FLX	NORFLX	FLX+NORFLX
**Q1**	73	102.5	208	69.5	98.75	201.5
**Q3**	155	186	328	156.25	184.25	306
**M-1SD**	57.97	85.28	167.49	62.51	78.88	160.67
**M+1SD**	191.29	218.24	385.31	181.19	223.57	385.52

Abbreviations: FLX, fluoxetine; FLX+NORFLX, active moiety; M-1SD/M+1SD, mean minus/plus 1 standard deviation; NORFLX, norfluoxetine; Q1, first interquartile; Q3, third interquartile. The unit of all values is ng/mL.

## Data Availability

The data presented in this study are available on request from the corresponding author. The data are not publicly available because these are sensitive health patient data.

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
