# Peer review of "Therapeutic Drug Monitoring in Children and Adolescents: Findings on Fluoxetine from the TDM-VIGIL Trial"

_pharmaceutics, 2023, doi:10.3390/pharmaceutics15092202_

Round 1
Reviewer 1 Report
The authors conducted therapeutic fluoxetine monitoring in children and adolescents in a large naturalistic-observational prospective multicenter clinical trial (“TDM-VIGIL”). Although the study seems to be an extension of authors previous work but on a large scale, the study still interesting and comprehensive.
I recommend the publication of the manuscript in its current form
Minor typographical mistakes have been fixed
Author Response
We kindly thank the reviewer for the positive evaluation of our manuscript.
We also attentively re-read the whole manuscript and adjusted some tiny stylistic and punctuation issues.
Reviewer 2 Report
This research strengthens the existing evidence that there is a direct correlation between the dosage of fluoxetine and the levels of fluoxetine, norfluoxetine, and the active component in the bloodstream of children and adolescents. This article is well-written and explained beautifully. However, as the conclusion mentions, more scientific research is required to support all the claims. This study focuses on the relationship between dosage and the response to fluoxetine, building upon previous discoveries. Conducting scientific experiments could further enhance the value and impact of this type of research. It would be great if the authors could provide more details about the HPLC experiments and some data. The authors are also suggested to enlighten about the possible toxicity associated with the higher dosages and their relationship with ages.
Author Response
We greatly appreciate the reviewer’s efforts in providing constructive comments, which have undoubtedly strengthened the quality of our work. Please find below the detailed answers and related additions made throughout the manuscript text.
Point-by-point responses:
1. “It would be great if the authors could provide more details about the HPLC experiments and some data.”
Thank you for this important advice. The corresponding information is now included in the Supplement S2.
“Supplement S2
Therapeutic Drug Monitoring of Fluoxetine and Norfluoxetine – Laboratory methods
Therapeutic drug monitoring (TDM) was performed according to the guidelines of the Working Group on Neuropsychopharmacology and Pharmacopsychiatry (AGNP). Blood collection from cubital veins in 7.5-ml monovettes without anticoagulants and additives took place at a steady-state trough level (after at least 14 days of consistent fluoxetine dosage; between 10 and 24 hours after the last dose) before the first daily drug intake. The blood was centrifuged at 1800 × g for 10 minutes and analyzed immediately (samples from Wuerzburg) or within a few days after postage to the TDM laboratory in Wuerzburg. Analysis of serum samples demonstrated that fluoxetine/norfluoxetine is not degraded under the following storage conditions: samples at +4°C in refrigerator for up to 1 week of storage and, for a longer period, at -20°C.
Fluoxetine serum concentrations were analyzed by an automated column-switching method coupled to an isocratic high-performance liquid chromatography system and a variable ultraviolet detector (Agilent 1200 Series; Agilent, Waldbronn, Germany). Chemicals and solvents of the highest level of purity and sertraline for calibration were purchased commercially from Sigma–Aldrich Chemie GmbH (Taufkirchen, Germany). A different washing and analytical eluent, different columns, and a different emission wavelength were used to achieve better separation of both substances from potential comedications.
The cycle of operation started with the injection of a 100-ml serum sample onto the extraction column (PerfectBond Vorsäulenkartusche 10×4, 0 mm CN 20 mm; MZ-Analysentechnik, Mainz, Germany) using a washing eluent of 10% (v/v) acetonitrile in deionized water at a flow rate of 1.25 ml/min. After 4 minutes, the electric six-port valve switched and the second pump transported the sample by back-flush mode onto the analytical column (MN-EC 150/4.6 NUCLEDUR 100-3 CN RP; Macherey–Nagel, Düren, Germany). The mobile phase contained 45% (v/v) acetonitrile and 10 mmol/l K2HPO4 in deionized water adjusted to pH 6.3 with H3HPO4. The flow rate was 1.25 ml/min at +30°C. The retention time was for fluoxetine 8.8 min and for norfluoxetine 8.1 min. The switching valve was reset after 14 minutes.
The emission wavelength of the ultraviolet detector was set at 210 nm. Data acquisition and integration were performed by means of the HP ChemStation (Agilent, Waldbronn, Germany). The absolute extraction recovery for fluoxetine was 93% and for norfluoxetine was 97%. The intra-assay coefficients of variation determined for both from 10 analyses (133.5 and 534 ng/ml) were in general less than 1%. The interassay variability for the analyte was in general less than 2.4%. The lower limit of quantification for both was 10.0 ng/ml. The method was linear in a range of 10–1335 ng/ml (R2 = 0.99).”
2. “The authors are also suggested to enlighten about the possible toxicity associated with the higher dosages and their relationship with ages.”
Thank you for this important advice to elaborate on possible toxicity associated with higher medication dose and in relation to age. This issue has now been addressed in the Introduction as follows (lines 151-161):
“Regarding the toxicity of fluoxetine, case reports of intoxications demonstrate relatively low toxicity compared to tricyclic antidepressants. Postmortem studies and case reports indicate values of 1300–7000 ng/ml for fluoxetine serum levels and 400-4000 ng/ml for norfluoxetine in fatal overdoses [25,26]. For adults, a laboratory alert value of 1000 ng/ml for the active moiety is reported, which is twice the upper limit of the therapeutic reference range [15]. Overdosing may lead to adverse effects such as tachycardia or drowsiness. This can in some cases already occur at approximately 400 ng/ml for the active moiety with significant interindividual variation [27-29]. In minors it appears, that the dosage per kg body weight is the most relevant factor. Females and adolescents (compared to males and adults, respectively), seem to be at higher risk for adverse effects of antidepressants, including SSRIs [30]. “
In addition to the above responses, we also attentively re-read the whole manuscript and adjusted some tiny stylistic and punctuation issues, which are marked in the text.

Reviewer 3 Report
The paper deals with an interesting, large pharmacology study aimed at determining fluoxetine therapeutic drug monitoring targets. It is based on a large population, much wider than those formerly used for the same goal. Albeit the analytical choice of HPLC can be considered a limitation, since LC-MS is today probably a better technique. However, the choice of HPLC can be considered a hystorically more robust option. Population traits have been pointed out and examined. Weakenesses of the present study and future perspectives in evaluating genetic polimorfisms, etc... have been addressed. The paper can therefore be considered well written, ready and interesting for varius clinician, from pharmacologists to mental health specialists.
Author Response
We kindly thank the reviewer for the positive evaluation of our manuscript.
Reviewer 4 Report
In this paper, "Therapeutic drug monitoring in children and adolescents: Findings on fluoxetine from the TDM-VIGIL Trial", the authors examined the relationship between doses and concentrations of fluoxetine and its active metabolite and also provided reference ranges for these active components (fluoxetine and active metabolite) in minors. The authors are to be congratulated on the impressive work provided in this study. However, there are several significant concerns and questions regarding the background of the study, interpretation of the data, incorrect citation, etc. The reviewer suggests that the manuscript should be thoroughly revised to improve the quality of the manuscript.
[Major comments]
1. The authors MUST provide correct information regarding the measurement method of fluoxetine and norfluoxetine because the reference cited (No. 33) was the paper for the measurement method of "sertraline". In addition, the quantification method of norfluoxetine was not described in the manuscript; the authors stated that "The concentrations of fluoxetine were determined using…" (lines 221-225). The mean concentrations of those active moieties shown in Table 1 are given to two decimal places, i.e., ten pg/mL increments. The data for linearity, the lower limit of quantification, accuracy, and precision of the analytical method, both within- and between runs, should be reported to ensure the reliability of the data.
2. The authors argued the effect of CYP2D6 genotypes on the pharmacokinetics of fluoxetine in lines 140-147 and 432-435. I was wondering why the authors decided not to perform CYP2D6 genotyping in this study. Please clarify the reason.
3. As the authors stated in the Introduction (lines 120-122), there are inter-individual variations in the clearance of fluoxetine (t1/2 of 4-6 days) and norfluoxetine (t1/2 of 4-15 days), probably due to the autoinhibition of CYP2D6 by fluoxetine itself (lines 401-403). At the same time, the authors defined a steady state as "after at least five half-lives" (lines 186-187). How did you assess whether the steady state is reached for each patient? I could not find this information even after reading the article cited as reference No. 34.
4. Due to the lack of data in the Results sections (from 3.2 to 3.6), I could not assess whether the interpretation of the data described in the manuscript was correct.
5. The authors mentioned the importance of medication adherence for the interpretation of this type of data analysis (lines 534-536). The reviewer agrees that information on medication adherence would be critical to assess the exposure-response relationship and to explore the target concentration range. Since the authors collected data on medication adherence (lines 212-214), why don't you analyze the effect of medication adherence on efficacy or concentration?
[Minor comments]
1. The abbreviation should be defined when the word is first displayed; for example, FLX should be defined on line 89 and NORFLX on line 117.
2. Regarding "Age" in Table 1, was the difference in decimal places essential for the interpretation of the results or further analysis? The age should be shown as an integer if this makes no sense to do so.
3. Please clarify the difference between the terms "reference range (in Abstract)" and "target range (in Discussion, line 436)".
4. Please provide the pre-defined rating scheme to rate the non-compliance (lines 212-214) as supplemental material. It would be helpful for readers to understand how medication adherence was assessed in this study.
5. To ensure the reliability of the quantification results, the authors should provide information on the stability of the analytes in human serum and the duration of storage of the samples after arrival in the Method section.
6. Regarding the second paragraph of the Introduction section (lines 109-116), I could not understand what to learn from this paragraph. Please clarify the authors' message.
7. The different aims of the study are described in the Introduction (lines 164-166) and the Discussion (lines 379-380). Please check for consistency throughout the manuscript.
Author Response
We greatly appreciate the reviewer’s efforts in providing detailed and constructive comments, which have undoubtedly strengthened the quality of our work. Please find below our detailed point-by-point responses. We also refer to changes and additions made throughout the manuscript and in the newly added supplementary materials.
Point-by-point responses
[Major comments]
- The authors MUST provide correct information regarding the measurement method of fluoxetine and norfluoxetine because the reference cited (No. 33) was the paper for the measurement method of “sertraline”. In addition, the quantification method of norfluoxetine was not described in the manuscript; the authors stated that “The concentrations of fluoxetine were determined using…” (lines 221-225). The mean concentrations of those active moieties shown in Table 1 are given to two decimal places, i.e., ten pg/mL increments. The data for linearity, the lower limit of quantification, accuracy, and precision of the analytical method, both within- and between runs, should be reported to ensure the reliability of the data.
We greatly apologize for the unintended omissions present in the first manuscript version and thank you for your important remarks. We have now complemented the necessary information in two places, first in the manuscript and second, in the newly added supplement.
- We have amended the following sentence (lines 253 – 256) of the manuscript: “The concentrations of fluoxetine and norfluoxetine were determined using an isocratic reversed-phase high-performance liquid chromatography (RP-HPLC) method with UV-absorbance detection on an Agilent 1200 series system (Agilent Technologies Inc., Santa Clara, CA, USA), as described in detail in the Supplement S2.”
- In order to provide a comprehensive description of measurement procedures for fluoxetine and norfluoxetine, we have integrated this information in the Supplement:
“Supplement S2
Therapeutic Drug Monitoring of Fluoxetine and Norfluoxetine – Laboratory methods
Therapeutic drug monitoring (TDM) was performed according to the guidelines of the Working Group on Neuropsychopharmacology and Pharmacopsychiatry (AGNP). Blood collection from cubital veins in 7.5-ml monovettes without anticoagulants and additives took place at a steady-state trough level (after at least 14 days of consistent fluoxetine dosage; between 10 and 24 hours after the last dose) before the first daily drug intake. The blood was centrifuged at 1800 × g for 10 minutes and analyzed immediately (samples from Wuerzburg) or within a few days after postage to the TDM laboratory in Wuerzburg. Analysis of serum samples demonstrated that fluoxetine/norfluoxetine is not degraded under the following storage conditions: samples at +4°C in refrigerator for up to 1 week of storage and, for a longer period, at -20°C.
Fluoxetine serum concentrations were analyzed by an automated column-switching method coupled to an isocratic high-performance liquid chromatography system and a variable ultraviolet detector (Agilent 1200 Series; Agilent, Waldbronn, Germany). Chemicals and solvents of the highest level of purity and sertraline for calibration were purchased commercially from Sigma–Aldrich Chemie GmbH (Taufkirchen, Germany). A different washing and analytical eluent, different columns, and a different emission wavelength were used to achieve better separation of both substances from potential comedications.
The cycle of operation started with the Injection of a 100-ml serum sample onto the extraction column (PerfectBond Vorsäulenkartusche 10×4, 0 mm CN 20 mm; MZ-Analysentechnik, Mainz, Germany) using a washing eluent of 10% (v/v) acetonitrile in deionized water at a flow rate of 1.25 ml/min. After 4 minutes, the electric six-port valve switched and the second pump transported the sample by back-flush mode onto the analytical column (MN-EC 150/4.6 NUCLEDUR 100-3 CN RP; Macherey–Nagel, Düren, Germany). The mobile phase contained 45% (v/v) acetonitrile and 10 mmol/l K2HPO4 in deionized water adjusted to pH 6.3 with H3HPO4. The flow rate was 1.25 ml/min at +30°C. The retention time was for fluoxetine 8.8 min and for norfluoxetine 8.1 min. The switching valve was reset after 14 minutes.
The emission wavelength of the ultraviolet detector was set at 210 nm. Data acquisition and integration were performed by means of the HP ChemStation (Agilent, Waldbronn, Germany). The absolute extraction recovery for fluoxetine was 93% and for norfluoxetine was 97%. The intra-assay coefficients of variation determined for both from 10 analyses (133.5 and 534 ng/ml) were in general less than 1%. The interassay variability for the analyte was in general less than 2.4%. The lower limit of quantification for both was 10.0 ng/ml. The method was linear in a range of 10–1335 ng/ml (R2 = 0.99).”
-
The authors argued the effect of CYP2D6 genotypes on the pharmacokinetics of fluoxetine in lines 140-147 and 432-435. I was wondering why the authors decided not to perform CYP2D6 genotyping in this study. Please clarify the reason.
We ourselves very much regret that we do not have genotyping data, as the study was part of a pharmacovigilance trial and primarily designed to capture adverse reactions and not genetic information. Thus, genotyping was unfortunately not part of the study protocol. We agree with the reviewer that a future study making use of genetic data in this context would be a fascinating and important research endeavor. -
As the authors stated in the Introduction (lines 120-122), there are inter-individual variations in the clearance of fluoxetine (t1/2 of 4-6 days) and norfluoxetine (t1/2 of 4-15 days), probably due to the autoinhibition of CYP2D6 by fluoxetine itself (lines 401-403). At the same time, the authors defined a steady state as “after at least five half-lives” (lines 186-187). How did you assess whether the steady state is reached for each patient? I could not find this information even after reading the article cited as reference No. 34.
This is an important point, and we agree that the information on steady-state levels has not been presented sufficiently in the text. We have therefore added the following clarification to line 202:
“All dosing steps of the medication were recorded with date in the medication protocol. The steady state (after at least five half-lives) was confirmed by the treating physicians and noted in the patient’s file.”
Supplement S2 adds that the steady state is reached “…after at least 14 days of consistent fluoxetine use." This criterion was available to the treating physicians.
-
Due to the lack of data in the Results sections (from 3.2 to 3.6), I could not assess whether the interpretation of the data described in the manuscript was correct.
Thank you for this remark. We are not entirely sure what specific information is being requested in this point, and thus, we apologize upfront for any potential misunderstandings. The results sections report the inferential statistics of the conducted tests, i.e., linear regressions, multiple linear regressions, cumulative link models, and receiver operating curve analysis. Given the fact that we analyzed both the last data point dataset and the all data point dataset, and did this separately for fluoxetine, norfluoxetine, and active moiety concentrations, numerous tests were performed. While this broad analysis requires some appropriate concision in reporting, we believe that the sections still include all primary statistical information, and we also made the maximal use of the space available by combining tables and descriptive sections explaining result directionality and significance. We hope that this clarification is to your satisfaction, but we are happy to elaborate further on this point, otherwise. -
The authors mentioned the importance of medication adherence for the interpretation of this type of data analysis (lines 534-536). The reviewer agrees that information on medication adherence would be critical to assess the exposure-response relationship and to explore the target concentration range. Since the authors collected data on medication adherence (lines 212-214), why don't you analyze the effect of medication adherence on efficacy or concentration?
Thank you for providing this valuable feedback. Non-adherence was an exclusion criterion and is mentioned in the paragraph on quality control criteria (line 323). One of our main aims was to calculate a reference range for responders, and therefore, it was important to exclude non-adherence. Specifically, we included participants classified as "certain" or "probable" based on the adherence rating scheme (please see below our response to the point 4 [minor comments] on how adherence was assessed). It would certainly be an interesting question to analyze the influence of adherence on serum levels and efficacy. However, only 2.5% of the collected visit time points did not fall into the "certain" or "probable" group. Thus, too few data points were available for analysis. Furthermore, the group of non-adherent participants is typically very heterogeneous, ranging from no adherence to occasional adherence and occasional forgetting. These reasons make in our opinion the comparability difficult.
“Supplement S1
Assessment of adherence to medication:
The following input options were available to treating physicians to assess adherence and were stored in the database.
- certain
- probably
- uncertain
- demonstrable intake error
- not yet known“
[Minor comments]
- The abbreviation should be defined when the word is first displayed; for example, FLX should be defined on line 89 and NORFLX on line 117.
We apologize for this omission. We have now added the abbreviations at their first occurrence in the text.
- Regarding "Age" in Table 1, was the difference in decimal places essential for the interpretation of the results or further analysis? The age should be shown as an integer if this makes no sense to do so.
Thank you very much for this feedback. Our “age” variable was calculated based on the birth date and investigation visit date; therefore, a decimal value was used. Nevertheless, the use of decimal values certainly did not play a significant role in the analysis. While we wish to keep our original analyses and we have marked this information in the table notes (“*age was entered in the analyses with decimal values”) for accuracy, we now also follow your recommendation for reporting, i.e., age (and age range) is now presented in the tables and elsewhere as an integer.
- Please clarify the difference between the terms "reference range (in Abstract)" and "target range (in Discussion, line 436)".
Thank you for pointing this out. We have used these two terms synonymously. For better comprehensibility, we have now replaced the term "target range" with "reference range" in the appropriate places.
- Please provide the pre-defined rating scheme to rate the non-compliance (lines 212-214) as supplemental material. It would be helpful for readers to understand how medication adherence was assessed in this study.
This is good advice, and such information will certainly help the readers to understand our adherence assessment. Accordingly, we have included a supplementary file with the following content:
“Supplement S1
Assessment of adherence to medication:
The following input options were available to treating physicians to assess adherence and were stored in the database.
- certain
- probably
- uncertain
- demonstrable intake error
- not yet known“
- To ensure the reliability of the quantification results, the authors should provide information on the stability of the analytes in human serum and the duration of storage of the samples after arrival in the Method section.
Thank you for this important advice. The corresponding information is now included in Supplement S2.
- Regarding the second paragraph of the Introduction section (lines 109-116), I could not understand what to learn from this paragraph. Please clarify the authors' message.
Thank you for your remark. We have re-examined the section noted by the reviewer. The first part concerning the receptor targets of fluoxetine is in our opinion important to explain the mechanisms for its overall better tolerability. This also accounts for its frequent use and serves as background information for the results regarding the frequency of side effects. In the second part, pharmacokinetic aspects are described, which seemed important to us as additional background information on the investigated substance. Even though these influencing factors were not specifically pursued in our study, they play a role in the interpretation of the results (e.g., influence of protein binding on the half-life, concentration required for efficacy, etc.). Therefore, we believe it is helpful to retain this paragraph to provide a comprehensive description of the investigated substance.
- The different aims of the study are described in the Introduction (lines 164-166) and the Discussion (lines 379-380). Please check for consistency throughout the manuscript.
Thank you for this very attentive observation. We have reworded the sentence in the Introduction (lines 180–184) to ensure consistency regarding the study objectives described in the Discussion and other parts of the manuscript:
“Therefore, the present naturalistic multicentric study was undertaken to investigate the relationships between dosage, serum concentration and its predictors in children and adolescents treated with fluoxetine. Additionally, our goal was to identify a preliminary age- and indication-specific therapeutic range of the active moiety, for which we explored serum level associations with clinical effects and adverse effects.”
In addition to the above responses, we also attentively re-read the whole manuscript and adjusted some tiny stylistic and punctuation issues, which are marked in the text.

Round 2
Reviewer 4 Report
I have read the revised manuscript with great interest.
Thank you for your honest responses.
The authors have almost adequately addressed my comments on the original version of the manuscript.
However, some points should be considered to improve the quality and readability of the manuscript.
1) Please recheck the manuscript to see if the abbreviations have been used appropriately throughout the manuscript. I have found many terms, "fluoxetine" and "norfluoxetine" in the revised manuscript after the definition of the abbreviations for them. In addition, the abbreviation "RP-HPLC" (line 239) should be omitted because this term was not presented after the definition.
2)In Supplement S2, the authors describe: "The working cycle started with the injection of a 100 ml serum sample onto the extraction column...". The serum volume needs to be corrected.
3) I recommend that the mean concentrations of active moieties reported in Table 1 not include decimals, i.e., pg/mL increments, due to the low sensitivity of the analytical method. This means that concentrations of the order of pg/mL determined by the method are not considered reliable.
4) A criterion to assess the achievement of steady-state had to be discussed. A minimum of 3.3 times the t1/2 is required to reach a steady state. According to the information in the manuscript (t1/2 of norfluoxetine ranges from 4-15 days, line 122), 14 days would not be sufficient to reach a steady state.
Author Response
First and foremost, we wish to convey our sincere appreciation for your time and dedication in reviewing our manuscript. Your current and previous comments have brought our attention to relevant aspects of the study and significantly enhanced the quality of our work. Please find below the answers to your remarks.
1) Please recheck the manuscript to see if the abbreviations have been used appropriately throughout the manuscript. I have found many terms, "fluoxetine" and "norfluoxetine" in the revised manuscript after the definition of the abbreviations for them. In addition, the abbreviation "RP-HPLC" (line 239) should be omitted because this term was not presented after the definition.
Thank you very much for this feedback. We apologize for not being coherent in handling abbreviations in our previous revision round. We have now replaced the terms with the introduced abbreviations and have also omitted the “RP-HPLC”, “PET” and “EMA” abbreviation appearing only once in the text.
2) In Supplement S2, the authors describe: "The working cycle started with the injection of a 100 ml serum sample onto the extraction column...". The serum volume needs to be corrected.
Thank you for this important feedback. We admit that the reported unit of serum volume was an error, which we have now corrected as follows: “…started with the injection of a 100 µl serum sample onto the extraction column...".
3) I recommend that the mean concentrations of active moieties reported in Table 1 not include decimals, i.e., pg/mL increments, due to the low sensitivity of the analytical method. This means that concentrations of the order of pg/mL determined by the method are not considered reliable.
We are happy to implement this helpful remark. We have now omitted the decimals in Table 1 for the active moiety, FLX and NORFLX.
4) A criterion to assess the achievement of steady-state had to be discussed. A minimum of 3.3 times the t1/2 is required to reach a steady state. According to the information in the manuscript (t1/2 of norfluoxetine ranges from 4-15 days, line 122), 14 days would not be sufficient to reach a steady state.
Thank you very much for your valuable comment. We acknowledge that the choice of a 14-day period represents a lower limit for determining the steady state. The value was derived by multiplying the lower boundary of the half-life range for fluoxetine and norfluoxetine (4 days) by a factor of 3.3 half-lives (which corresponds to reaching 90% of steady state concentration at constant dosing), and then rounding up. Our multicenter study was embedded within naturalistic clinical settings. This decision was made because an early determination of serum levels is required in clinical practice, particularly to promptly identify poor metabolizers, if necessary. In addition to examining the relationship between dose and serum concentration, we also aimed to investigate associations with clinical effects and adverse reactions. Adopting a longer time window until the first level measurement could obscure possible associations, especially taking into account evidence for clinical improvement within the first two weeks of taking antidepressants (including fluoxetine) (Tsujii et al., 2022).
Given the above, we have made the following amendment in the revised manuscript (line 249): "All dosing steps were recorded along with corresponding dates in the medication protocol. The steady state (after at least 3.3 half-lives in relation to the lower range limit) was confirmed by the treating physicians and documented in the patient's medical records."
We have also added the following limitation (lines 723-727): “Sixth, due to clinical practicability, the prompt detection of potential poor metabolizers, and the capture of serum levels for early treatment effects within the first two weeks, we opted for the earliest possible time point as steady state [65]. This is important to note that this choice could potentially result in reference ranges that are rather underestimated.”
References
- Tsujii, T.; Sakurai, H.; Takeuchi, H.; Suzuki, T.; Mimura, M.; Uchida, H. Predictors of response to pharmacotherapy in children and adolescents with psychiatric disorders: A combined post hoc analysis of four clinical trial data. Neuropsychopharmacology Reports 2022, 42, 516-520.